# Nanocrystalline TiO_2_ Sensitive Layer for Plasmonic Hydrogen Sensing

**DOI:** 10.3390/nano10081490

**Published:** 2020-07-29

**Authors:** Enrico Gazzola, Michela Cittadini, Marco Angiola, Laura Brigo, Massimo Guglielmi, Filippo Romanato, Alessandro Martucci

**Affiliations:** 1Department of Physics and Astronomy “G. Galilei”, University of Padova, Via Marzolo 8, 35131 Padova, Italy; gazzola.lann@gmail.com (E.G.); filippo.romanato@unipd.it (F.R.); 2Department of Industrial Engineering, University of Padova, Via Marzolo 9, 35131 Padova, Italy; michela.cittadini@gmail.com (M.C.); marcoangiola@gmail.com (M.A.); laurabrigo@gmail.com (L.B.); massimo.guglielmi@unipd.it (M.G.); 3Institute for Materials Manufacturing, CNR, Area Science Park, Strada Statale 14 Km 163.5, 34149 Trieste, Italy; 4Istituto di Fotonica e Nanotecnologie, CNR, via Trasea 7, 35131 Padova, Italy

**Keywords:** optical sensors, plasmonic sensors, hydrogen, titania, gold nanoparticles

## Abstract

Solution processed TiO_2_ anatase film was used as sensitive layer for H_2_ detection for two plasmonic sensor configurations: A grating-coupled surface plasmon resonance sensor and a localized surface plasmon resonance sensor with gold nanoparticles. The main purpose of this paper is to elucidate the different H_2_ response observed for the two types of sensors which can be explained considering the hydrogen dissociation taking place on TiO_2_ at high temperature and the photocatalytic activity of the gold nanoparticles.

## 1. Introduction

Plasmonic gas sensors are optical sensors that exploit either propagating or localized surface plasmons to transduce an analyte concentration into an optical signal. In the first case, a surface plasmon is excited at a metal-dielectric interface with a dimension much larger than the wavelength of the incoming light (for example, a gold film deposited on a glass substrate). This plasmonic mode is called surface plasmon polariton (SPP) and is an electromagnetic wave coupled to collective electron oscillations, propagating along the metal-dielectric interface. Since SPPs are extremely sensitive to changes in the optical properties of the surrounding materials, they can be exploited for developing high-performance sensing devices called Surface Plasmon Resonance sensors (SPR sensors) [1,2]. SPR sensors based on propagating plasmons have been successfully applied especially for biosensors [3,4,5], but also for gas and vapor pollutants [6,7,8], in particular using metal oxide as sensitive layer [9]. The second approach exploits gold nanoparticles (NPs), which exhibit strong extinction coefficients in the visible or NIR due to their ability to support a localized surface plasmon resonance (LSPR). LSPR-based devices are a well-established approach for fast, label-free biosensing [10,11]. The spectral position of the LSPR of Au NPs immersed in a metal oxide supporting matrix is sensitive to both electrons exchange at the gold/matrix interface as well as the permittivity variations of the matrix; both of these effects have been exploited for developing hydrogen and carbon monoxide optical gas sensors [12,13,14]. Since the air refractive index variation caused by the mere presence of hydrogen is hardly detectable, many authors investigated specific mechanisms involved in LSPR-based sensors response to hydrogen, while little attention was paid to the SPR sensors based on propagating plasmons. It has long been believed that H_2_ molecules can dissociate interacting with oxygen vacancies of TiO_2_ at 300 °C, releasing electrons which are transferred to the conduction band through intragap states. [15,16,17,18,19]. Buso et al. and Della Gaspera et al. concluded that LSPR changes from H_2_ reduction of TiO_2_-Au films at high temperature (above 250 °C) and of TiO_2_-Au-Pt films at room temperature are induced by charge transfer to the Au NPs [12,20]. This conclusion was supported by the work of Carpenter et al., where the LSPR shifts of Au NPs dispersed in an yttrium-stabilized-zirconia matrix was attributed to the same H_2_ interaction mechanism [21]. On the contrary, Larsson et al. attributed the H_2_ shift of the LSPR of Au NPS to the H_2_-induced dielectric changes of the nearby Pt NPs separated by a SiO_2_ support [22]. Plasmonic NPs themselves are believed to have photocatalytic properties, being able to induce H_2_ and O_2_ dissociation under visible light, either in presence of an oxide matrix or not [23,24,25]. An increasing amount of literature associates their photocatalytic action to the emission of hot electrons by the LSPR decay [26,27], which makes the gold NPs surface very reactive, inducing the dissociation of H_2_ also in the absence of any active material, like Pd or Pt [28] and under broadband incoherent illumination [29]. While some debate still exists about the exact mechanisms driving the hot electrons transfer and participation to the chemical reactions, their emission today is considered a key feature of LSPR-based photochemistry [26]. Some evidences suggest that the interaction of the Au NPs with dissociated hydrogen then leads to the formation of an AuH layer over the NPs surface [29], while the contribution of TiO_2_ active sites in H_2_ dissociation was found to be negligible in the experiments reporting this phenomenon [28,30]. On the other hand, Collins et al. recently demonstrated that hydrogen dissociation and spillover could directly happen on semiconducting oxides surfaces [31]. In the present paper, we employ both propagating and localized plasmons for developing two types of hydrogen sensor using TiO_2_ anatase films as sensitive layers, prepared from nanocrystalline ink deposited by spin coating. Scope of the present study is to elucidate the interaction mechanisms of H_2_ with the TiO_2_ sensitive layer which induce different responses for the two kinds of plasmonic sensors. For both sensors, we discuss the origin of the observed shift of the plasmonic resonances, which cannot be explained considering only the variation of the refractive index induced by the target gas filling the pore of the sensitive matrix. In the case of propagating plasmons sensor, a TiO_2_ film was deposited on a metallic grating, which can support SPPs. In the case of LSPR sensor, a sub-monolayer of Au NPs was coated with TiO_2_ film. We compared the two transducing platforms and the importance of the TiO_2_ matrix in each case, by discussing the role of the H_2_ dissociation mechanism and the influence of the operative temperature. Sensors with SiO_2_ in place of TiO_2_ or without any oxide layer were also tested for comparison.

## 2. Materials and Methods

### 2.1. Plasmonic Gratings Fabrication

Figure 1a shows the multilayer profile of a generic thin-film dielectric-metal-dielectric plasmonic structure, with a sinusoidal pattern acting as a grating coupler. Sinusoidal gratings are fabricated through soft lithographic techniques, starting from sinusoidal patterns realized by laser interference lithography (LIL) on Photoresist S1805 films (Microposit, Rohm and Haas Company, Philadelphia, PA, USA) deposited on silicon [32]. The fabrication process consists in the realization of negative replica made of polydimethylsiloxane (PDMS) of the nanostructured masters (some hundreds of nm pitch, tens of nanometers of peak-to-valley height), and in the pattern transfer by soft Nano Imprint Lithography (NIL) on a dielectric organic-inorganic hybrid sol–gel film (dielectric 1 in Figure 1a) [33]. The film is synthesized starting from phenyl-bridged silsesquioxanes (ph-PSQ), and spin-coated over fused silica substrates with a final thickness of 200 nm [34,35]. Ph-PSQ works as a thermoset resist for NIL, and is specifically selected to guarantee a reduced lateral and vertical shrinkage, allowing nanostructure preservation during high temperature (<700 °C) thermal treatments. PDMS replica were gently pressed with a finger on fresh-deposited ph-PSQ films, and the assembly was cured for 30 min at 80 °C in an oven, before delicately peeling the mold off the sample. After the pattern-transfer, the structure is hard baked at 500 °C for 30 min to promote successive thermal stability. A thin bimetallic film is evaporated over the patterned layer, consisting on 37 nm of silver, that provides good plasmonic properties, and 7 nm of gold to protect silver from oxidation [36]. The plasmonic gratings used for the present investigation have a period of 570 nm and a peak-to-valley height of 70 nm, measured by Atomic Force Microscopy (AFM-Park Europe, Hamburg, Germany) before the deposition of the TiO_2_ or SiO_2_ layer (see Figure 1b).

### 2.2. Au Monolayer Fabrication and TiO_2_ and SiO_2_ Film Deposition

As described in [37], sub monolayer Au NPs were deposited on fused silica glass starting from colloidal Au NPs dispersed in ethanol. After deposition, the monolayer was heat-treated in air at 400 °C for one hour.

For the TiO_2_ layer we used a solution of TiO_2_ NPs dispersed in methanol. As described in [38], TiO_2_ NPs were synthesized by adding 10.5 mmol of titanium(IV) isopropoxide (TTIP) dropwise into a previously prepared solution containing water (H_2_O), hydrochloric acid (HCl) and methanol (MeOH) with molar ratios H_2_O/TTIP = 12.25, HCl/TTIP = 1.72. The solution was stirred for 60 min at room temperature, heated in oil bath at 70 °C and kept at this temperature for 4 h under reflux. Particles were then precipitated and dispersed in a minimal amount of MeOH obtaining a clear TiO_2_ anatase sol.

For the SiO_2_ layer a standard solution of tetraethoxysilane (TEOS) in ethanol was used, containing TEOS:ethanol:H_2_O:HCl = 1:4:2:0.01 molar ratios. A solution of the silica precursors was first made with only 1/3 of the required ethanol and stirred for 30 min. Only prior to film deposition, the remaining ethanol was added to reach the nominal sol composition.

Thin films were deposited by spin coating at 3000 rpm for 30 s on the plasmonic gratings or on the Au sub monolayer. All the samples were dried at 100 °C for 5 min in air and finally annealed in a muffle furnace at 300 °C for 1 h in air. A characterization of the structural and optical properties of the bare Au sub monolayer, the bare TiO_2_ films and the Au sub monolayer coated with TiO_2_ or SiO_2_ has been already reported in [37,38,39], respectively. In particular, TiO_2_ nanoparticles with mean diameter of 4.0 ± 1.5 nm with anatase crystalline phase were observed by HRTEM and XRD measurements [38].

### 2.3. Optical Characterization and Gas Sensing Setup

The main instrument used for the optical characterization of our structures was a Variable Angle Spectroscopic Ellipsometer (V-VASE, J.A. Woollam Co., Lincoln, NE, USA). Transmittance at normal incidence and ellipsometry parameters Ψ and Δ were measured using the V-VASE, at two different angles of incidence (65°, 75°) in the wavelength range of 300–1500 nm. Optical constants n and k and films thicknesses were evaluated from Ψ, Δ and transmittance data using WVASE32 ellipsometry data analysis software, fitting the experimental data with Cauchy dispersion for non-absorbing region, Gaussian and Tauc-Lorentz oscillators for Au LSPR peak and UV absorption edge, respectively [40]. The thickness of the TiO_2_ films was estimated in 130 nm, with uncertainty of a few percent, mainly due to thickness non uniformity. Uncertainty associated to the ellipsometry measurement itself, in the investigated spot, is typically negligible in comparison.

The plasmonic properties of the nanostructured gratings were characterized using the same instrument. In this case, reflectance measurements were performed, with front side illumination and p-polarized light, as a function of the wavelength and at different incidence angles. Reflectance dips were observed in the spectra in correspondence to the resonant excitation of SPP modes. Sensing measurements on the plasmonic gratings with a TiO_2_ sensitive matrix were performed under direct illumination conditions, using a customized commercial cell (Heat Cell J. A. Woollam Co. Lincoln, NE, USA), equipped with fixed input and output windows perpendicular to the incoming and reflected beams, mounted over the ellipsometer sample holder. Measurements were carried out at a fixed angle of incidence of 70°. Reflectance spectra were recorded at 300 °C in a dry N_2_ environment, then under exposure to 1% and 5% H_2_, and finally again in a N_2_ atmosphere to regenerate the sensor, with flow rate of 0.4 L/min. Such measurements were repeated for a number of detection/regeneration cycles, each exposure step lasting few tens of minutes. The plasmonic dips were monitored and their shifts were correlated to the interaction with the analyte. Sensing tests on the Au monolayer films were performed under the same atmosphere and flux conditions, monitoring the LSPR absorption peak position by optical absorbance measurements on the wavelength range of 250–900 nm, using a Harrick gas flow cell (with 5.5 cm path length) coupled to a JascoV-570 spectrophotometer (Jasco, Easton, MD, USA). The substrate area was approximately 1 cm × 2 cm, and the spectrophotometer beam illuminated a 6 mm × 1.5 mm sample area at normal incidence. All the gas sensing measurements were repeated several times for checking the reproducibility of the observed shifts.

### 2.4. Plasmonic Grating Spectra Characterization

In principle, both interfaces of the metal layer with dielectric materials can support SPP modes: let us define the “front interface” as the directly illuminated one and “back interface” the opposite one. The metal layer is thin, i.e., of the order of the typical penetration depth of the SPP evanescent fields. Nevertheless, in a structure like ours, where the two surfaces are conformally patterned as depicted in Figure 1a, only the modes on the front interface can usually be excited [41], unless we deal with a symmetric structure. In that case the modes on the two identical interfaces would not exist independently; they would couple to generate thin-film modes instead, namely the Long-Range and Short-Range SPP modes [42,43,44]. Our structure is not manifestly symmetric, since the upper and lower materials (labelled dielectric 1 and 2, respectively, in Figure 1) are different. Their refractive index dispersion curves, measured by spectroscopic ellipsometry, are reported in Figure 2.

In spite of this, the structure may still be effectively symmetric at some wavelengths. Therefore, we need to characterize its symmetry in order to understand which kind of modes we will excite. In order to do this, we avail ourselves of an Effective Medium Approximation (EMA) approach, in which the contribution of consecutive layers to an effective permittivity is weighed by the SPP field exponential decay [45]. This means that, if *δ* is the SPP penetration depth in a semispace composed of a succession of layers in the z direction, the EMA dielectric constant is calculated as follows:(1)$$\varepsilon_{EMA}=\frac{2}{\delta }\left[\int_{0}^{d_{1}}\varepsilon_{1}e^{-2\frac{z}{\delta }}dz+\int_{d_{1}}^{d_{2}}\varepsilon_{2}e^{-2\frac{z}{\delta }}dz+\cdots \right]$$
where *d*_1_, *d*_2_, … are the layers thicknesses and *ε*_1_, *ε*_2_, … their permittivity. We calculated by means of this approach the *ε_EMA_* of both the dielectric and metallic semispaces probed by the front interface SPP. Since *δ* is itself a function of the effective permittivity, this iterative procedure was carried until convergence. We used the final values of the two effective permittivity to provide the SPP momentum through the well-known dispersion relation:(2)$$\beta =k_{0}\sqrt{\frac{\varepsilon_{EMA}\left(dielectric\right)\cdot \varepsilon_{EMA}\left(metal\right)}{\varepsilon_{EMA}\left(dielectric\right)+\varepsilon_{EMA}\left(metal\right)}}$$
where *k*_0_ is the incident light momentum. For comparison, we repeated the procedure for the back interface. The procedure was carried for a range of wavelengths *λ* to produce the curves reported in Figure 3. At the wavelengths where the two curves crosses, the structure is effectively symmetric; therefore, at these wavelengths we would expect not to observe a single plasmonic resonance with the calculated momentum *β*, but two resonances with two different momenta, slightly higher (the Short Range SPP) and lower (the Long Range SPP) than *β*. On the contrary, when the two curves are sufficiently separated, the two interfaces would be decoupled and only the upper could be excited; the observed resonances therefore would possess the momenta provided by the blue curve. Simulated spectra generated by the implemented Chandezon’s algorithm [46,47] confirmed the predictions of Figure 3 and provided the additional information that for wavelengths over 1200 nm the momentum mismatch is sufficiently high for the two interfaces to be effectively decoupled. With this tool, we are ready to characterize the experimental spectrum. A typical spectrum, collected at an incident angle of 70°, is reported in Figure 4 together with a corresponding simulated spectrum generated by Chandezon’s method. A set of dip is observed, with good agreement between experiment and simulation. By comparison with the momenta provided by the above procedure, the three dips can be associated to the resonant coupling between the incident radiation and front interface SPP modes. Given the grating lattice momentum, *G* = 2π/570 nm^−1^, and the momentum transferred by the incident light to the surface mode, *k_T_* = 2πsin(70°)/λ nm^−1^, the different coupling conditions for each resonance can be identified [48]. In detail, the SPP in the infrared is excited via subtractive coupling, i.e., $\beta =G-k_{T}$, and is therefore labelled SI−; on the contrary, the mode labelled SI+ is excited via additive coupling, $\beta =G+k_{T}$. At lower wavelengths, the spectrum presents a SPR dip identified as a second order additive excitation, $\beta =2G+k_{T}$, thus labelled SI_2_^+^. The corresponding dip in the experimental spectrum appears distorted and will not be exploited in the sensing tests.

## 3. Results

### Hydrogen Detection Experiments

We performed sensing experiments at 300 °C, as described in Section 2.3, by exposing each platform to either 1% or 5% H_2_ and monitoring the SPP dip shift in the case of the plasmonic grating sensor, or the LSPR shift in the case of Au monolayer sensor. Exposures to the analyte were repeated in a cyclical procedure, alternated to the flux of pure N_2_ to regenerate the sensors. With this procedure, repeatable and reversible shifts of the plasmonic resonances in response to the analyte can be observed, therefore suppressing the weight of random fluctuations in the sensitivity estimation. Tests on the TiO_2_-coated plasmonic gratings show that interaction with the target gas causes a blue-shift of the SPP dips; typical reflectance spectra are showed in Figure 5 and the measured shifts are reported in Table 1. Similarly, a blue-shift of the Au LSPR peak was observed in the sensing tests of the Au monolayer coated with TiO_2_ as can be seen in Figure 6a which shows the absorbance spectra and the Optical Absorption Change (OAC), defined as the difference between the absorbance in H_2_ and in N_2_ (OAC = Abs_H2_ − Abs_N2_), during H_2_ exposure.

The normalized absorbance at resonance as a function of time is also reported, in Figure 6b, to show the repeatable and reversible sensor response when cyclically exposed to H_2_ and N_2_. It is possible to appreciate a fast response time of 52 s (defined as the time needed to reach 90% of the total response) and a slower recovery time of 412 s (defined as the time needed to recover 90% of the baseline). Both kinds of structure exhibit a relevant response to the presence of hydrogen at 300 °C, indicating that the plasmonic modes can detect variations of the optical properties of the environment in their proximity.

However, the target gas has an index of refraction almost identical to pure N_2_, which cannot justify the sensor response. Therefore, it is necessary to look for other interaction mechanisms which could explain the magnitude of the observed shifts of the resonant wavelengths.

We performed some additional measurements for comparison, namely sensing tests at room temperature, at 300 °C without the TiO_2_ layer, and at 300 °C with SiO_2_ in place of the TiO_2_. These additional tests were performed for both propagating and localized plasmons sensors, providing further useful information. Both propagating and localized plasmons sensors showed no sensitivity at room temperature, suggesting that the involved mechanisms become effective at high temperature. Plasmonic gold gratings either without TiO_2_ film or with SiO_2_ film showed no response to the analyte, neither at room temperature nor at 300 °C. In summary, they responded to hydrogen only at 300 °C with the TiO_2_ matrix. On the contrary, LSPR-based sensors were sensitive to H_2_ at 300 °C both when TiO_2_ was absent (see Figure 6c) and when it was replaced by SiO_2_ matrix (see Figure 6d). The measured shifts are reported in Table 2.

In order to explain the different behavior between plasmonic grating sensor and Au monolayer sensor, we shall therefore acknowledge that more than one mechanism is involved which induce permittivity variations in the structures. Indeed, the situation appears easier for the grating-based sensors, where the results exclude an active role of the gold film and suggest a mechanism strictly related to the presence of the TiO_2_ film and effective only at high temperature (300 °C). In order to provide an independent measurement of the permittivity variation of the TiO_2_ film when the platform was exposed to H_2_, we also exploited spectroscopic ellipsometry. Flatness is required for a reliable spectroscopic ellipsometry; thus, it was performed on a TiO_2_ layer deposited over a flat gold film over a Si substrate. The measured variations were between −0.05 and −0.08, with a clear dependence on the wavelength, and are reported in Figure 7 (black curve). We also verified that the gold film alone (in absence of a TiO_2_ layer) exposed to H_2_ shows no detectable variation of its permittivity.

The variation of the dielectric constant of the active matrix can be related to a variation of the carrier concentration in the conduction band. In fact, it was already reported [8,9] that the dissociation of H_2_ on the TiO_2_ active sites can be the source of electrons that can be transferred into the conduction band inducing a variation of the conductivity and hence also a variation of the permittivity of the TiO_2_ matrix. Based on this assumption, we will estimate the required amount of charge carrier corresponding to the measured SPP dip shift and its plausibility under our experimental conditions, in the case of plasmonic grating sensor. Then we will discuss the case of Au monolayer sensor and the related LSPR shift. Figure 8 shows a conceptual scheme of the processes for the different structures.

## 4. Discussion

### 4.1. Model for Permittivity Variation

Since the presence of H_2_ cannot directly provide a detectable permittivity variation of the TiO_2_ matrix, we suppose that such effect is due to an injection of electrons into the matrix, which modifies the charge carrier density of the material and in turn its complex permittivity, $\varepsilon_{r}+i\varepsilon_{i}$. We therefore need a model to convert the observed SPP dip shifts into a variation of charge carrier density. First of all, we can ignore the variation of *ε_i_*, because it does not affect in a detectable way the spectral position of the resonance dip, but only slightly its Full Width at Half Maximum and its minimum intensity, according to our simulations and to literature [49]. We use the Drude–Lorentz model to relate the permittivity, *ε_r_*, to the charge carrier density, N, and the damping factor, *γ*, at a frequency *ω*, as follows [50,51]:(3)$$\varepsilon_{r}=\varepsilon_{b}-\frac{N}{\omega^{2}+\gamma^{2}}\frac{q_{e}^{2}}{\varepsilon_{0}m^{*}}$$
where *q_e_* is the electron charge, *ε*_0_ the vacuum permittivity, *m** the charge carrier effective mass and *ε_b_* the “background” dielectric constant, usually identified with the (real) permittivity at very high frequencies. In the optical regime, absorption is small and we can reasonably neglect *γ*^2^ with respect to *ω*^2^. Since we make differential measurements, *ε_b_* cancels out and we do not need to provide its actual value. We need to provide a value for the effective mass *m**, instead. The *m** values for TiO_2_ reported in literature, range from less than 1 to 100 times the electron mass, depending on the fabrication process and the actual structure of TiO_2_ [52]. For our specific case, namely TiO_2_ anatase of dimensions 5–10 nm, *m** is reported to be approximately 10 times the electron mass; therefore, we will use this value [53]. We can therefore correlate the carrier density variation δN to the permittivity variation *δε_r_* by using the following equation:(4)$$\delta \varepsilon_{r}=-\frac{\delta N}{\omega^{2}}\cdot \frac{q_{e}^{2}}{\varepsilon_{0}m^{*}}$$

We used computer simulations with Chandezon’s method to evaluate the permittivity variations required to achieve the experimentally observed SPP dips shifts for the plasmonic grating sensor, and then Equation (4) to determine the values of δN which could account for them. The measured shifts together with the corresponding permittivity variations and estimated charge density injection in the conduction band of the TiO_2_ matrix are summarized in Table 1. It is possible to note that the observed SPP dip shifts are consistent with an injection of less than 1 carriers/nm^3^ and in particular that the charge injection at higher wavelength would be much less than at lower wavelength. Since the TiO_2_ has a band-gap of about 3.4 eV [54], lying in the near ultraviolet region, we may speculate that this difference may be due to the higher energy of the incident photons, promoting more effectively the transition of electrons to the conduction band through the intragap states.

### 4.2. Model for H_2_-TiO_2_ Interaction

We now discuss the origin of the additional charge carriers. Our interpretation is based on the possibility that H_2_ molecules, when interacting with active sites in the TiO_2_ matrix at 300 °C, dissociate into H atoms, which then enter the lattice and ionize releasing electrons into the matrix. This process, schematically summarized in Figure 8a, was first suggested by Göpel et al. in 1983, based on a previous extensive investigation on the role of electrons injection in rutile [15]. The same process was assumed by other authors to be the fundamental reason for the sensitivity of TiO_2_ anatase to H_2_; more recently, Mukherjee et al. invoked the same mechanism to justify a change in the TiO_2_ resistance under hydrogen exposure [19]. According to these authors, the active sites for the dissociation process are provided by oxygen vacancies in the oxide matrix, which also induce the formation of intragap states, which in turn promote the charge transfer into the conduction band. This is qualitatively consistent with all our observations, since the absence of active sites in TiO_2_ at low temperature and in SiO_2_ at any temperature explains the effectiveness of our plasmonic grating sensor only for TiO_2_ at 300 °C.

We want also provide a rough quantitative evaluation of the consistency between the magnitude of the estimated charge injection and of the adsorbed hydrogen during the exposure time. Since each dissociated H_2_ molecule would release two electrons, the possibility of less than 1 dissociation/nm^3^ to occur would be sufficient to account for any measured shift. Our evaluation includes two steps: an estimation of the active site density in the TiO_2_ matrix and a count of the H_2_ molecules crossing the interaction volume during the exposure time. In order to estimate the density of active sites, we start from the TiO_2_ anatase density, that is, 3 g/cm^3^, according to previously reported measurements [55]. Since its molar mass is 79.866 g/mol, it is easily converted in a density number of TiO_2_ molecules: ρ_Ti_ = 22.6 molecules/nm^3^. The active sites for the H_2_ dissociation are supposed to be oxygen vacancies of the form Ti^3+^; the fraction of TiO_2_ molecules containing an active site is estimated to be around 2%, which would imply a density number of active sites of almost 0.5 sites/nm^3^ [56,57]. Now, we evaluate the number of available hydrogen molecules. Given the H_2_ concentration of 50000 ppm in nitrogen, its density number ρ_H_ is 5% of the nitrogen density number, which gives, under standard ambient temperature and pressure, ρ_H_ = 0.013 molecules/nm^3^. Since our flux is driven by a pressure drop of 1 atm, we may assume an approximately double density, ρ_H_ = 0.02 molecules/nm^3^. This is an instant value; we shall also consider the integrated flow during the exposure time. Given the sensor response time, of about 1 min, and the gas flow rate of 0.4 L/min, we conclude that the about η_H_ = 10^21^ molecules flow through the cell during the exposure time. Of course, not all these molecules will actually interact with the sensitive layer, while entering the cell, moving in the turbulent flow and possibly finding the way out. As a rough estimation we can simply divide η_H_ by the cell volume of 4 cm^3^, obtaining approximately 2 molecules/nm^3^. In the case of 1% H_2_ concentration, the concentration is 5 times less. The estimated concentration of adsorbed H_2_ molecules is consistent with our numerical results and therefore the interpretation of the H_2_-TiO_2_ interaction mechanism at 300 °C seems to be correct.

### 4.3. Discussion of the Au Monolayer Sensor

In the case of the Au monolayer sensor, we observed a LSPR peak shift induced by H_2_ exposure, not only in the presence of the TiO_2_ layer, but also in the case of SiO_2_ and even with bare Au NPs (see Figure 6). Therefore, it is clear that further effects come into play in the presence of Au NPs, at high temperature. Plasmonic nanostructures are known to exhibit photocatalytic behavior, even in the absence of an oxide substrate [26,27] and under low intensity visible light [29,58]. It was also demonstrated that, in contrast to semiconductor photocatalysts, plasmonic photocatalysts increase their quantum efficiency with temperature [59,60]. To explain the photochemical interaction of H_2_ with Au NPs without any oxide matrix, hot electrons emission is often invoked [26,27]. They would induce H_2_ dissociation on the Au surface [28] with consequent formation of an AuH layer, resulting in a decrease of the medium effective refractive index and consequently a blue-shift of the LSPR [29]. It is reported that the dissociation rate is reduced in presence of TiO_2_, due to the flow of hot electrons from the metal to the semiconductor, which prevents them to participate to the hydrogen dissociation process, while SiO_2_ has a positive effect because, on the contrary, it does not drain electrons [30]. We may therefore speculate that in the case of the Au monolayer sensor we can have a plasmonic photocatalytic effect, as represented in Figure 8b, possibly related to the hot-electrons emission by the LSPR decay or to a catalytic effect induced by the low dimension of the Au NPs. The absence of any response in the grating structures under the same conditions suggests that this effect is related to the low 3D dimension of the Au NPs. We did not observe an enhancement of hydrogen sensitivity when the NPs are embedded in SiO_2_, while we did observe an improved sensitivity when they are embedded in the TiO_2_ matrix. Indeed, we are led to assume that the optical response of the Au monolayer coated with TiO_2_ is the result of an interplay between different effects. As depicted in Figure 8c, we shall assume that both the H_2_ dissociation on the TiO_2_ active sites, previously described for the plasmonic grating sensor, and the H_2_ dissociation on the Au NPs, induced by plasmonic photocatalytic effect are present, lacking any reason to expect any of them to be suppressed. As showed, both effects result in a blue-shift, which explains the improved sensitivity of the Au monolayer sensor with respect to plasmonic grating sensor. We should consider the possibility of additional effects due to the interaction of the two. For example, if an electron draining phenomenon like the one described in [30] was present, it would affect the reference spectrum, because electrons would flow into the oxide also in the absence of H_2_. Further investigations are needed to unravel the interplay between these two effects in the Au monolayer coated with TiO_2_. What we can state is that the application of the TiO_2_ active layer is necessary in the case of gold grating-based sensors, because the SPP mode does not provide photocatalytic activity on its own, in contrast with the LSPR sensor.

## 5. Conclusions

We fabricated, characterized and tested two different kinds of TiO_2_-based plasmonic sensors for hydrogen detection, obtained from solution processed TiO_2_ film. The observed SPR shift was attributed to an electron transfer effect due to H_2_ molecules dissociation on TiO_2_ at 300 °C, considering that the effect of a simple pores filling is negligible. We demonstrated also that the Au-NPs monolayers exhibit photocatalytic activity inducing hydrogen dissociation, allowing the Au monolayer sensor to be sensitive also in the absence of an oxide layer. Based on our experimental evidence and the proposed interaction mechanism, we concluded that the TiO_2_-mediated hydrogen dissociation is crucial in the grating-based SPR sensor, because the SPP supported by the gold grating does not exhibit photocatalytic activity.

## Figures and Tables

**Figure 1 nanomaterials-10-01490-f001:**
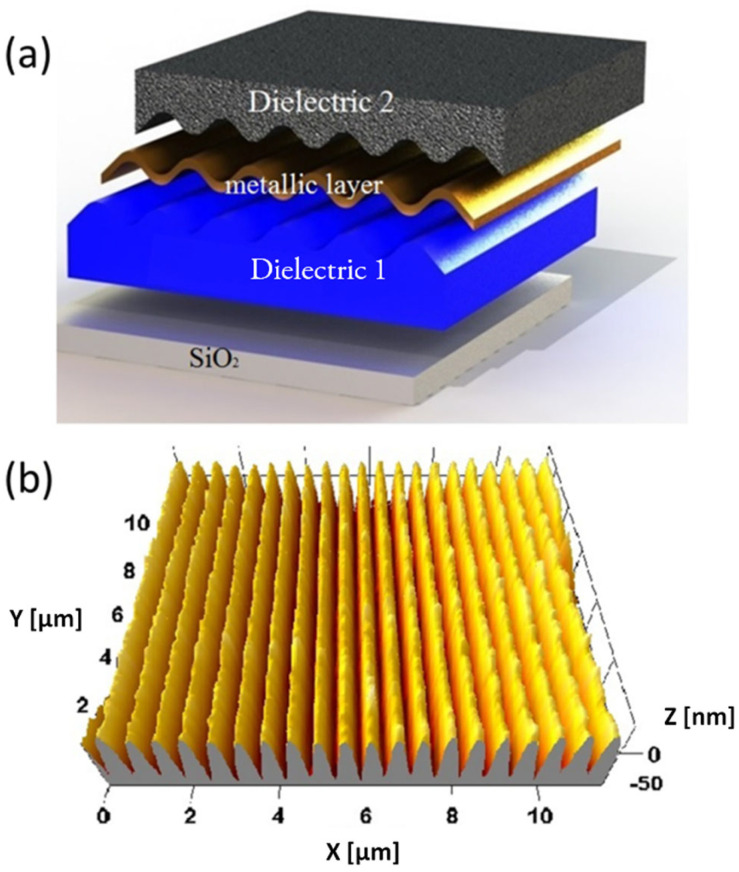
(**a**) sketch of the plasmonic crystal multilayer profile; it is formed by the SiO_2_ glass substrate, the phenyl-bridged silsesquioxanes (ph-PSQ) films with the patterned grating (dielectric 1) and the gold layer. On top of th metallic layer it is possible to deposit a sensitive film (dielectric 2) that can be TiO_2_ or SiO_2_ as described in the text. (**b**) three-dimensional Atomic Force Microscopy (AFM) reconstruction of the grating surface, before the deposition of dielectric 2.

**Figure 2 nanomaterials-10-01490-f002:**
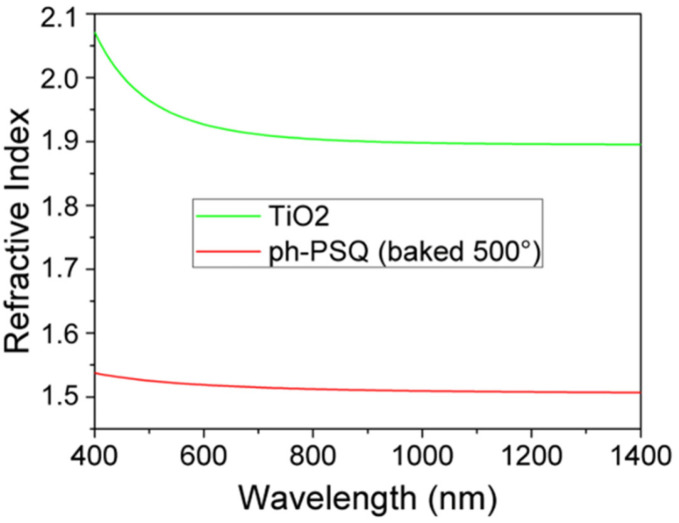
Refractive indices of the two functional materials used for developing the plasmonic grating sensor: the green curve refers to the TiO_2_ sensitive layer (dielectric 2 of Figure 1a) and the red one to the organic-inorganic hybrid sol-gel SiO_2_-based resist (dielectric 1 of Figure 1a).

**Figure 3 nanomaterials-10-01490-f003:**
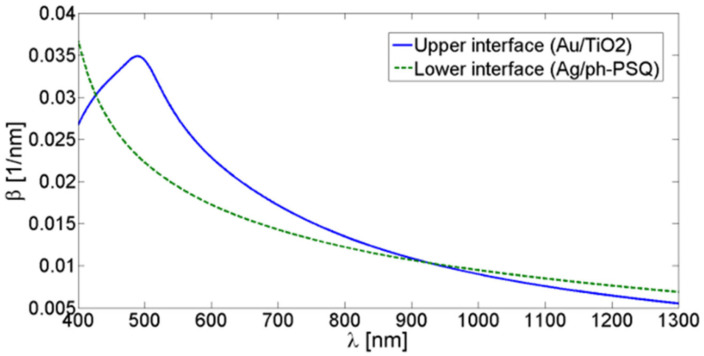
Estimated momenta of the surface plasmon polaritons (SPPs) at the front and back interfaces, as a function of the incident wavelength in the range of interest.

**Figure 4 nanomaterials-10-01490-f004:**
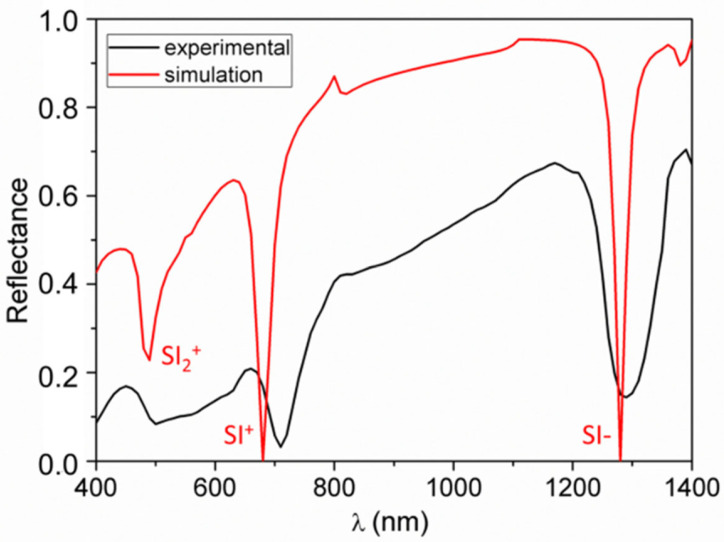
Experimental reflectance spectrum of the plasmonic grating (black curve) compared to simulation (red curve). The illumination is front-side at a fixed incidence angle of 70°. The nature of the modes is identified by comparison to their theoretical momenta, as follows: the label SI stands for single-interface mode, with a sign indicating if the coupling is additive or subtractive and a number “2” for a mode coupled through the second diffraction order.

**Figure 5 nanomaterials-10-01490-f005:**
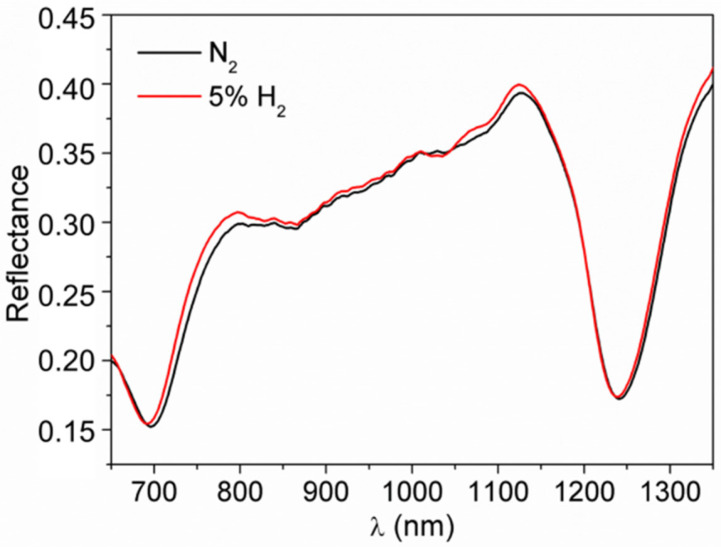
Sensing test at 300 °C towards 5% H_2_ for plasmonic grating coated with TiO_2_.

**Figure 6 nanomaterials-10-01490-f006:**
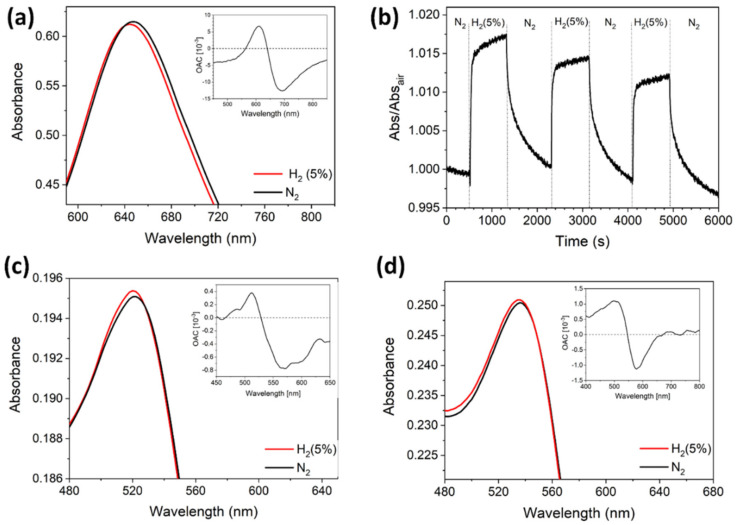
Gas sensing measurements on Au monolayer samples exposed to 5% H_2_ at 300 °C. (**a**) Au monolayer coated with TiO_2_; (**b**) dynamic sensing at 610 nm for a repeated N_2_-H_2_-N_2_ cycle for the Au monolayer coated with TiO_2_; (**c**) bare Au monolayer, (**d**) Au monolayer coated with SiO_2_. The insets show the Optical Absorbance Change (OAC) defined as the difference between the absorbance in H_2_ and in N_2_ (OAC = Abs_H2_ − Abs_N2_).

**Figure 7 nanomaterials-10-01490-f007:**
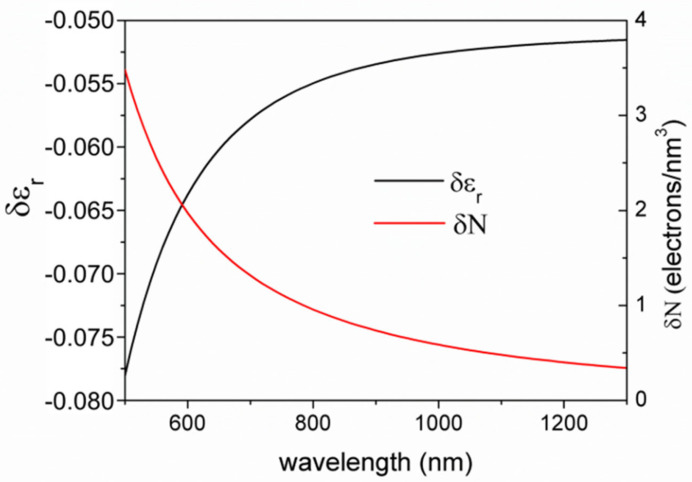
Black curve: variation of the relative permittivity of the TiO_2_ layer exposed to 5% hydrogen, directly measured by spectroscopic ellipsometry; Red curve: the corresponding charge carrier injection in the TiO_2_ conduction band, estimated by inverting Equation (4).

**Figure 8 nanomaterials-10-01490-f008:**
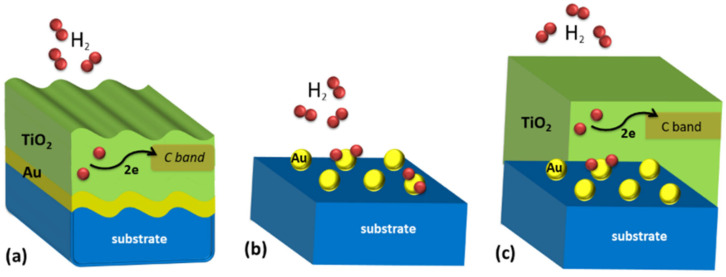
Schemes of the possible processes responsible of H_2_ gas sensing response. (**a**) In the grating-based sensor at 300 °C operative temperature, the H_2_ dissociates on TiO_2_ active sites and electrons are released into the TiO_2_ Conduction Band. (**b**) For LSPR based sensors made of Au NPs at 300 °C operative temperature it is possible to have H_2_ dissociation directly on the Au NPs. (**c**) When the TiO_2_ layer is deposited over the Au NPs monolayer, it is possible to have both H_2_ dissociation on TiO_2_ active sites and on the Au NPs.

**Table 1 nanomaterials-10-01490-t001:** Plasmonic grating sensor: observed resonance shifts for the two monitored SPP dips in 1% and 5% H_2_ atmosphere, with corresponding permittivity variation (*δε*) according to simulations and charge carrier injection in the conduction band of TiO_2_ (δN) estimated with Equation (4).

SPP Dip [nm]	H_2_	Shift [nm]	δε_r_	δN [nm^−3^]
700 nm	1%	−2.9 ± 0.1	−0.03	0.6
700 nm	5%	−4.5 ± 0.2	−0.045	1
1240 nm	1%	−0.9 ± 0.2	−0.015	0.1
1240 nm	5%	−2.2 ± 0.1	−0.04	0.3

**Table 2 nanomaterials-10-01490-t002:** Au monolayer sensor: observed Au localized surface plasmon resonance (LSPR) shift in 5% H_2_ atmosphere.

Structure	LSPR Position in N_2_ [nm]	LSPR Position in 5% H_2_ [nm]	Shift [nm]
Au	521.1 ± 0.1	519.7 ± 0.1	−1.4 ± 0.2
Au-SiO_2_	536.0 ± 0.1	534.9 ± 0.1	−1.1 ± 0.1
Au-TiO_2_	647.9 ± 0.1	645.0 ± 0.1	−2.9 ± 0.2

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
