# Peer review of "Nanocrystalline TiO2 Sensitive Layer for Plasmonic Hydrogen Sensing"

_nanomaterials, 2020, doi:10.3390/nano10081490_

Round 1

Reviewer 1 Report

This paper details the construction and characterization of LSPR-based sensors capable of detecting hydrogen in the presence of nitrogen. Overall, this was a good paper, though several questions that the authors can answer would be appreciated. Furthermore, while it is appreciated that English is not the authors' first language, some additional editing for English style would be appreciated. It is recommended that this paper be published with major revisions, though the answers to some of the questions may make these minor revisions.

  1. Figure 6b and accompanying text: Why is there a significant recovery time? Is this indicative of a dissociated hydrogen atoms sticking strongly to the gold nanoparticles? In addition, was there a hydrogen concentration study performed? Would increasing the hydrogen concentration cause a saturation of the gold nanoparticles and reduce or eliminate the sensitivity?
  2. Was there a thickness study of the TiO2 layer? Previous work on other hydrogen sensors using TiO2 (usually some form of catalytic effect) have shown a significant effect on the performance based on the thickness of TiO2.
  3. Page 11, Lines 372-373: The correct reference needs to be added.
  4. Page 12, Line 396: The correct reference needs to be added.

Author Response

We thank the reviewers for useful comments. All the points raised by the reviewers were addressed and the point by point rebuttal is reported hereafter. The whole manuscript was revised improving the English style.

This paper details the construction and characterization of LSPR-based sensors capable of detecting hydrogen in the presence of nitrogen. Overall, this was a good paper, though several questions that the authors can answer would be appreciated. Furthermore, while it is appreciated that English is not the authors' first language, some additional editing for English style would be appreciated. It is recommended that this paper be published with major revisions, though the answers to some of the questions may make these minor revisions.

  1. Figure 6b and accompanying text: Why is there a significant recovery time? Is this indicative of a dissociated hydrogen atoms sticking strongly to the gold nanoparticles? In addition, was there a hydrogen concentration study performed? Would increasing the hydrogen concentration cause a saturation of the gold nanoparticles and reduce or eliminate the sensitivity?

The main scope of the paper is to compare and provide an interpretation of the different the gas sensing response of two plasmonic sensor configurations: a grating-coupled surface plasmon resonance sensor and a localized surface plasmon resonance sensor with gold nanoparticles. For both sensors we used the same sensitive layer (nanocrystalline TiO2 film) and we investigate the origin of the different behaviors showed by the two types of sensors. In our previous papers (ref. 7,8,9,12,20,31) we studied the sensitivity and selectivity of the two sensors, performing several experiments using different target gas, sensing materials and operative conditions. Regarding the H2 interaction with Au nanoparticles, we did an extensive study in ref 31, and we demonstrated the importance of Pt to improve the H2 dissociation.

  1. Was there a thickness study of the TiO2 layer? Previous work on other hydrogen sensors using TiO2 (usually some form of catalytic effect) have shown a significant effect on the performance based on the thickness of TiO2.

In this paper we did not study the effect of the TiO2 layer because it was out of the scope of the paper, as pointed out in point 2. In a previous study (see for example J. Mater. Chem., 2011, 21, 4293) we demonstrated that film thickness and porosity are key factors for sensor performances.

  1. Page 11, Lines 372-373: The correct reference needs to be added.

The ref was added.

  1. Page 12, Line 396: The correct reference needs to be added.

The ref was added.

Reviewer 2 Report

The manuscript presents Nanocrystalline TiO2 sensitive layer for plasmonic hydrogen sensing. The following comments should be answered before the consideration of publication.

  1. There is no schematic for the sensor experimental set up.
  2. There is no data on selectivity of the sensor without which it cannot be ascertained that the sensor will sense H2 only.
  3. There is no mention of the limit of detection of the sensor.
  4. No comment on the reproducibility, i.e., whether there were many devices tested showing similar behavior.
  5. There is no table to summarise the results of previous H2 sensors and to compare those with the current study without which it is not possible for the reader to realise the importance of this work.
  6. The following references are recommended to be added in the manuscript.

[1] IEEE Transactions on Electron Devices, vol. 63, no. 1, pp. 476-480, January, 2016.

[2] IEEE Sensors Journal, vol. 13, no. 6, pp. 2423-2427, April 2013.

Author Response

We thank the reviewers for useful comments. All the points raised by the reviewers were addressed and the point by point rebuttal is reported hereafter. The whole manuscript was revised improving the English style.

The manuscript presents Nanocrystalline TiO2 sensitive layer for plasmonic hydrogen sensing. The following comments should be answered before the consideration of publication.

  1. There is no schematic for the sensor experimental set up.

A detailed description of the experimental set-up was reported in our previous publications, see for example ref 8 and 12.

  1. There is no data on selectivity of the sensor without which it cannot be ascertained that the sensor will sense H2 only.

The main scope of the paper is to compare the gas sensing response of two plasmonic sensor configurations: a grating-coupled surface plasmon resonance sensor and a localized surface plasmon resonance sensor with gold nanoparticles. For both sensors we used the same sensitive layer (nanocrystalline TiO2 film) and we investigate the origin of the different behaviors showed by the two types of sensors. In our previous papers (ref. 7,8,9,12,20,31) we studied the sensitivity and selectivity of the two sensors, performing several experiments using different target gas, sensing materials and operative conditions.

  1. There is no mention of the limit of detection of the sensor.

See answer to question 2, in previous paper we estimated a detection limit of 0.01% H2.

  1. No comment on the reproducibility, i.e., whether there were many devices tested showing similar behavior.

As reported in the experimental section 2.3, all the gas sensing measurements were repeated several times for checking the reproducibility of the observed shifts.

  1. There is no table to summarise the results of previous H2 sensors and to compare those with the current study without which it is not possible for the reader to realise the importance of this work.

The present work focuses on the discussion of the sensing mechanisms involved in the two sensor configurations, rather than a comparison with the already published H2 gas sensors. As stated in the answer to question 2, we studied the sensitivity and selectivity of the two sensors in previous publications.

  1. The following references are recommended to be added in the manuscript.

[1] IEEE Transactions on Electron Devices, vol. 63, no. 1, pp. 476-480, January, 2016.

[2] IEEE Sensors Journal, vol. 13, no. 6, pp. 2423-2427, April 2013.

The two references are not related to H2 gas sensor and regard conductometric gas sensor, not optical gas sensor, so it seems that they are not so relevant for the paper.

Round 2

Reviewer 1 Report

Thank you for making the suggestions and answering my questions. I recommend this paper be published in its current form.

Author Response

We thank the reviewer for the positive response.

Reviewer 2 Report

  1. Figure 1 is not clear. Please provide clear one.
  2. XRD analysis of TiO2 shoule be added in the manuscript.
  3. A table should be added in the manuscript to summarise the results of previous H2 sensors and to compare those with the current study without which it is not possible for the reader to realise the importance of this work.

Author Response

1. Figure 1 is not clear. Please provide clear one.

Figure 1 was revised and the caption was modified to make more clear the sketch of the plasmonic grating.

2. XRD analysis of TiO2 should be added in the manuscript.

As reported in the section 2.2 the characterization of the structural and optical properties of the bare Au sub monolayer, the bare TiO2 films and the Au sub monolayer coated with TiO2 or SiO2 has been already reported in [37], [38] and [39]. In particular XRD and HRTEM of the TiO2 nanocrystals are reported in ref 38, showing that TiO2 nanoparticles have a mean diameter of 4 nm and anatase crystal structure. A phrase has been added in section 2.2 to clarify this point.

3. A table should be added in the manuscript to summarize the results of previous H2 sensors and to compare those with the current study without which it is not possible for the reader to realize the importance of this work.

We already answered to this point in our previous rebuttal. We remark that the importance of this work is not in such a comparison. We focused on exploring the mechanisms involved in the interaction between plasmonic sensors and hydrogen molecules and their different roles in the two different kind of sensors. It is not a review about H2 sensors performances, neither the presentation of sensors optimized for competitive purpose, therefore the required comparison would be an irrelevant addition for the scope of the paper. To better clarify the importance of the present work we modified both the abstract and the introduction pointing out the main purpose of the paper.